# Deep Eutectic Solvent-Based Microextraction of Lead(II) Traces from Water and Aqueous Extracts before FAAS Measurements

**DOI:** 10.3390/molecules25204794

**Published:** 2020-10-19

**Authors:** Mohamed A. Habila, Najla AlMasoud, Taghrid S. Alomar, Zeid A. AlOthman, Erkan Yilmaz, Mustafa Soylak

**Affiliations:** 1Advanced Materials Research Chair, Department of Chemistry, College of Science, King Saud University, P.O. Box 2455, Riyadh 11451, Saudi Arabia; zaothman@ksu.edu.sa; 2Department of Chemistry, College of Science, King Saud University, P.O. Box 2455, Riyadh 11451, Saudi Arabia; 3Department of Chemistry, College of Science, Princess Nourah bint Abdulrahman University, Riyadh 11671, Saudi Arabia; nsalmasoud@pnu.edu.sa; 4Department of Analytical Chemistry, Faculty of Pharmacy, Erciyes University, 38039 Kayseri, Turkey; erkanyilmaz@erciyes.edu.tr; 5Nanotechnology Research and Application Center (ERNAM), Erciyes University, 38039 Kayseri, Turkey; 6Department of Chemistry, Faculty of Sciences, Erciyes University, 38039 Kayseri, Turkey; msoylak@gmail.com

**Keywords:** deep eutectic solvent, microextraction, lead(II), water, atomic absorption spectroscopy

## Abstract

Microextraction procedures for the separation of Pb(II) from water and food samples extracts were developed. A deep eutectic solvent composed of α-benzoin oxime and iron(III) chloride dissolved in phenol was applied as a phase separator support. In addition, this deep eutectic mixture worked as an efficient extractor of Pb(II). The developed microextraction process showed a high ability to tolerate the common coexisting ions in the real samples. The optimum conditions for quantitative recoveries of Pb(II) from aqueous extracts were at pH 2.0, conducted by adding 150 µL from the deep eutectic solvent. The quantitative recoveries were obtained with various initial sample volumes up to 30 mL. Limits of detection and limits of quantification of 0.008 and 0.025 µg L^−1^ were achieved with a relative standard deviation (RSD%) of 2.9, which indicates the accuracy and sensitivity of the developed procedure. Recoveries from the reference materials, including TMDA 64.2, TMDA 53.3, and NCSDC-73349, were 100%, 97%, and 102%, respectively. Real samples, such as tap, lake, and river water, as well as food samples, including salted peanuts, chickpeas, roasted yellow corn, pistachios, and almonds, were successfully applied for Pb(II) analysis by atomic absorption spectroscopy (AAS) after applying the developed deep eutectic solvent-based microextraction procedures.

## 1. Introduction

Lead in the environment has increased due to urbanization and industrial development [1,2,3]. Lead may contaminate the environment via mining centers, dust movement, industrial byproducts, and exhaust from contaminated gasoline. In addition, accumulation and transference through the food chain leads to a more significant spread into soil, plants, water, and food, making its way into the human body [4,5]. Common pathways that lead to human exposure to lead include the respiratory system, where approximately 30–40% of the lead is transferred to the blood via absorption, which results in the circulation of the lead. The possibility of the circulating lead binding with erythrocytes is high, leading to its rapid distribution throughout the body, including in organs such as the liver and brain, and in the bones [6]. In addition, the presence of lead in the human body causes oxidative stress through reactive oxygen species production as well as reducing antioxidant defenses’ ability such as by affecting superoxide dismutase and catalase enzymes [7,8,9,10,11,12,13,14]. Therefore, the accurate determination of lead is an important issue in analytical chemistry because of the very low concentrations in some samples as well as the presence of a matrix in real samples, which may interfere with instrumental determination [15,16,17]. However, the application of sample pretreatment, including preconcentration techniques such as solid-phase extraction [18,19,20,21], solid-phase microextraction [22,23], and dispersive liquid–liquid microextraction [16,24,25], has led to significant improvements in analytical methods and enabled the accurate determination of the targeted pollutants. The microextraction-based preconcentration process is more promising than traditional extraction processes such as solid-phase extraction or liquid-phase extraction [26,27]. Solid-phase microextraction and liquid-phase microextraction were recently introduced, and have many advantages, such as fast operation, as well as requiring a lower amount of chemicals during their conduction [28]. However, the liquid-phase microextraction process is superior to solid-phase microextraction due to its simple and quick phase separation as well as its ability to apply a direct injection of the separated phase for instrumental analysis [27,29]. The development of the microextraction steps to preconcentrate Pb(II) has been reported; for example, Zhou et al. [24] applied dithizone to chelate Pb(II) for dispersive liquid liquid microextraction (DLLME) application with a precision of 2.12% (RSD, *n* = 7) and a detection limit of 0.95 ng L^−1^. Mandlate et al. [30] determined Pb(II) in soft drinks by applying dispersive liquid‒liquid microextraction (DLLME) with GF-AAS and reported LODs of 0.072 ng L^−1^. Mohammedi et al. [31] applied a ligandless-based dispersive liquid–liquid microextraction process for Pb(II) microextraction from water for flame atomic absorption spectrometry determination. Faraji and Helalizadeh [16] determined Pb(II) in urine by applying DLLME before ultraviolet and visible spectrophotometry with a linear range from 0.01 to 100 μg L^−1^ and relative standard deviations of 15.3%. Afzali et al. [32] developed displacement-dispersive liquid–liquid microextraction by solidification of floating organic drops for preconcentration and determination of Pb(II) by flame atomic absorption spectrometry with a linear range between 4 and 700 ng mL^−1^ with a detection limit of 0.7 ng mL^−1^. Shah et al. [33] developed an in-syringe system using ionic liquid for microextraction of Pb(II) with an LOD of 0.281 μg L^−1^. Recently, a deep eutectic solvent was applied for microextraction as green solvents; it was composed of the coordination of hydrogen bond donors, such as long-chain carboxylic acids with hydrogen bond acceptors, such as quaternary ammonium salts or choline chloride, which reduce the melting point of the mixture, forming an organic viscus solution that is used to assess the separation of the aqueous and organic phases during microextraction procedures [34,35,36,37,38]. For example, Abolghasemi et al. [39] used microextraction procedures with a deep eutectic solvent for the preconcentration of pesticides from food. Sorouraddin et al. [40] applied the deep eutectic solvent for the development of reversed-phase-based microextraction for the separation of cadmium and zinc. Zounr et al. [41] extracted lead from water and food samples by applying deep eutectic and air-assisted liquid microextraction, and achieved LODs of 0.60 ng L^−1^. Alavi et al. [42] analyzed Pb(II) in blood samples using electrothermal atomic absorption spectroscopy (ETAAS), after applying deep eutectic microextraction procedures containing 1-octanol and *N,N,N*-cetyltrimethyl ammonium bromide as an extractor with a linear range between 1 and 200 ng mL^−1^. However, deep eutectic solvents have a broader composition, which may include metal chloride associations with electron donor molecular compounds; further investigations for microextraction application are required.

This work aimed to develop a microextraction process by applying a mixture of deep eutectic solvents, composed of α-benzoin oxime and iron(III) chloride dissolved in phenol, for the attraction and separation of Pb(II) from aqueous samples. The parameters affecting the accuracy and sensitivity of the microextraction procedures were optimized, as well as the real samples’ application, and the tolerance ability of foreign ions was investigated.

## 2. Results and Discussion

### 2.1. Investigations of the pH of the Sample Solution on the Recovery of Pb(II)

Deep eutectic solvents are promising for the attraction of heavy metals from aqueous solutions because they are entirely composed of nitrogen and oxygen in their structure, in addition to the main skeleton of organic derived materials including carbon and hydrogen [34,35]. The separation and extraction of heavy metal ions from a sample solution depends on the pH of the medium due to the complexation between metal ions and the ligand [23,43]. For this optimization, the pH was increased to 7. The recovery was calculated and is presented in Figure 1. A strongly acidic medium of pH 2 showed the best recovery values, while weakly acidic mediums of pH 5 and 6 had lower recovery.

### 2.2. Optimization of the Deep Eutectic Solvent Quantity

In the developed microextraction procedures, the deep eutectic mixture was formed by mixing the α-benzoin oxime and iron(III) chloride dissolved in phenol, resulting in a sticky solution that could be uniformly distributed in the sample solution under vortex shaking conditions. This led to the formation of a homogenous cloudy solution; at this stage, Pb(II) was collected from the aqueous solution and attached to the deep eutectic organic phase due to the presence of α-benzoin oxime. The advantage of the prepared deep eutectic solvent is the presence of the three components together, including phenol and iron(II) chloride, which are important for the deep eutectic solvent structure as well as the nitrogen atom and hydroxyl group in the structure of the α-benzoin oxime ligand, which enhance the ability of the deep eutectic mixture to coordinate lead(II) from aqueous samples; this is a key point in the developed microextraction process. The role of phenol is to act as donor for the electrons (hydrogen bond donors (HBDs)) and the iron(III) chloride is to act as hydrogen-bond acceptors (HBAs)) during formation of the deep eutectic solvents. Phenol is reported for application as HBDs with choline chloride as HBAs [44]. The iron(III) chloride is reported as HBAs with thiomalic acid [45], ethylene glycol glycerol, malonic acid, pentaerythritol, xylitol, serine, alanine and glycine as HBDs [46]. In the present work, the combination between phenol and iron(III) chloride is applied for the first time based on our survey for preconcentration purposes. The deep eutectic solvents facilitate the phase separation during the microextraction process. The separation of the deep eutectic solvent phase by centrifugation led to an effective microextraction process. The volume of the deep eutectic mixture was a key parameter in achieving the quantitative recoveries. Therefore, it was investigated and is shown in Figure 2. The effect of changing the volume of the deep eutectic solvents was correlated to the microextraction efficiency in terms of recovery. The volume of deep eutectic solvent in the range of 50–100 µL exhibited a recovery of approximately 80%, while the complete microextraction with the best recovery was approximately 100%, which was achieved by injecting at least 150 µL of the deep eutectic solution mixture.

### 2.3. Tolerance of Foreign Ions in Relation to Recovery

During real applications, the conditions of the microextraction media were affected by the presence of coexisting ions that may compete with the targeted analyte, leading to a reduction in the sensitivity of the method and a reduction in the recovery [47,48,49,50]. The coexisting ions, such as F^−^, PO_4_^3−^, Ni^2+^, NO_3_^−^, Co^2+^, Zn^2+^, CO_3_^2−^, K^+^/Cl^−^, Mg^2+^, Ca^2+^, SO_4_^2−^, and Fe^2+^, were investigated using the developed DES-ME process for Pb(II) preconcentration to assess the selectivity and tolerance. The recoveries were in the range between 94% and 103% (Table 1), indicating that the developed DES-ME process for Pb(II) is matrix-independent.

### 2.4. Analytical Merit for the Developed DES-ME of Pb(II)

The sample volume is an important factor for obtaining a high preconcentration factor [51,52,53,54,55,56,57,58]. The sample volume was studied in the range from 10 to 45 mL (Figure 3). The developed DES-ME of Pb(II) exhibited a high recovery up to 30 mL. By considering the last extracted volume for the Pb(II) solution with an instrumental injection of 500 µL, the preconcentration factor was 60, as calculated via Equation (1):(1)$$Preconcentration\;Factor=\frac{Maximum\;Sample\;volume\;\left(mL\right)}{Last\;extracted\;volume\;\left(mL\right)}.$$

The developed DES-ME of Pb(II) exhibited limits of detection (LODs) and limits of quantification (LOQs) of 0.008 and 0.025 µg·L^−1^, respectively, with an RSD% of 2.9, which indicates accuracy and sensitivity. In addition, the recoveries from the reference materials, including TMDA 64.2, TMDA 53.3, and NCSDC-73349, were 100%, 97%, and 102%, respectively (Table 2).

### 2.5. Real Food and Water Samples Analyses after Applying the Developed Deep Eutectic-Based Microextraction of Pb(II)

Environmental monitoring for the determination of Pb(II) in daily consumed samples is recommended to ensure low levels of Pb(II) contamination and control the quality of food and water [47,48,49,50,51,52,53]. A group of real samples, including various food and water samples, were applied for the developed DES-ME of Pb(II) separation and preconcentration before atomic absorption spectroscopy AAS analysis. Table 3 presents the levels of detected Pb(II) in the analyzed samples. In view of the obtained results, it can be concluded that Pb(II) was detected only in trace amounts, which indicates a very low contamination level; in addition, it reveals the accuracy of the developed DES-ME of Pb(II), which was also applicable in the case of the samples with various matrices.

Furthermore, the addition/recovery experiments were conducted for the evaluation of the performance of the developed DES-ME of Pb(II) under various analyte concentration levels and in the presence of different real sample matrices. Table 4 presents the recovered Pb(II) concentrations with various spiking levels, which are in agreement with the added one, with recovery between 92% and 101%.

The efficiency of the developed DES-ME of Pb(II) was compared with other microextraction methods from the literature [21,23,24,30,33,34,42]. The limits of detection of 0.008 µg L^−1^ for the developed DES-ME were between those of the other methods, as shown in Table 5.

## 3. Materials and Methods

### 3.1. Reagents and Chemicals

Deionized water was used for the preparation and dilution of all solutions. The chemicals applied in this work were all of analytical grade. A Pb(II) stock solution of 1000 mg L^−1^ was prepared from the nitrate salt and diluted daily for designed experiments. α-Benzoin oxime, iron(III) chloride, phenol, sodium fluoride, sodium phosphate, nickel(II) nitrate, hexahydrate, sodium nitrate, lead(II) nitrate, cobalt(II) nitrate, zinc(II) nitrate, sodium carbonate, potassium chloride, magnesium nitrate hexahydrate, calcium chloride, sodium sulfate, and iron(II) nitrate were purchased from Sigma (St. Louis, MO, USA). Sodium hydroxide, hydrochloric acid, nitric acid, and hydrogen peroxide were purchased from Merck (Darmstadt, Germany).

### 3.2. DES-ME of Pb(II)

The deep eutectic extraction solution was obtained by mixing iron(III) chloride with phenol at a molar ratio of 1:5, warming the mixture at 50 °C for 10 min, and stirring it for 60 min at room temperature. Afterwards, 0.01 g of α-benzoin oxime was added to the mixture, which was stirred for an additional 60 min at room temperature, forming a homogenous sticky solution that was applied directly for the microextraction process. The melting point was measured by a Shimadzu (Kyoto, Japan) DSC-60 and found to be 28.6 °C. In addition, the viscosity of the DES mixture was measured by Viscolite 700 and found to be 47 cps. The samples containing Pb(II) in a 50 mL tube were adjusted to pH 2 using a Sartorius PT-10 pH meter (Sartorius, Gottingen, Germany). The pH was adjusted and stabilized using a phosphate buffer and controlled to the desired pH by adding drops of diluted hydrochloric acid (0.01 M) or sodium hydroxide (0.01 M). Then, a portion of the deep eutectic extraction solution (approximately 150 µL) was injected. The mixture was vortexed for 5 min to form a cloudy mixture. Centrifugation for 12 min at 4000 rpm was applied for phase separation. The aqueous phase was discarded, and the remaining phase, including the Pb(II) analyte, was dissolved in nitric acid, to a final volume of 500 μL, which was detected by a flame atomic absorption spectrometer (Perkin Elmer model 3110, PerkinElmer, Inc., Shelton, CT, USA). The recovery was calculated by Equation (2):(2)$$Recovery\%=\left(\frac{C_{f}}{C_{0}}\right)\;\times \;100,$$
where *C_f_* is the final concentration, and *C_0_* is the initial concentration.

The microextraction experiments were conducted in three replications. The Excel Microsoft Office program was used to calculate the average and standard deviation of Pb(II) concentration measurements for the replicates. The standard errors were calculated from the Equation (3):(3)$$\mathrm{Standard}\;\mathrm{errors}\;\left(\mathrm{SE}\right)=\frac{\mathrm{Standard}\;\mathrm{deviation}\;}{\surd n}$$
where n is the number of replicates

### 3.3. Validation and Real Applications

Standard certified reference materials, including TMDA 64.2, TMDA 53.3, and NCSDC-73349, were applied for evaluating the performance and efficiency of the developed deep eutectic-based Pb(II) microextraction. In addition, tap, lake, and river water samples were filtered through a 0.45-μm pore size filter, and then, 10 mL were applied for the developed deep eutectic-based Pb(II) microextraction following the above procedures. For food samples, including NCSDC-73349, salted peanuts, chickpeas, roasted yellow corn, pistachios, and almonds, 1.0 g was prepared, using nitric acid digestion to obtain a clear solution that was adjusted to 10 mL with deionized water. Two milliliters of the food sample extract were applied to the developed deep eutectic-based Pb(II) microextraction. A spiking of a certain amount of Pb(II) and the recovery from tap water, chickpeas, and pistachios was investigated using Pb(II) concentrations of 0.5, 1.0, and 2.0 mg L^−1^.

## 4. Conclusions

The deep eutectic solvent mixture composed of α-benzoin oxime and iron(III) chloride dissolved in phenol showed a high efficiency for interaction and association with Pb(II) during the microextraction process. In addition, the deep eutectic mixture played an important role in the phase separation, leading to an efficient and selective preconcentration with high recovery from aqueous solutions. The relative standard deviation of 2.9 reveals the stability of the results and indicates the suitability of the developed deep eutectic microextraction for real applications. The addition/recovery analysis showed close agreement between the spiked and detected concentrations, with recoveries between 92% and 106%. The developed DES-ME of Pb(II) exhibited high tolerance for various ions, including F^−^, PO_4_^3−^, Ni^2+^, NO_3_^−^, Co^2+^, Zn^2+^, CO_3_^2−^, K^+^/Cl^−^, Mg^2+^, Ca^2+^, SO_4_^2−^, and Fe^2+^.

## Figures and Tables

**Figure 1 molecules-25-04794-f001:**
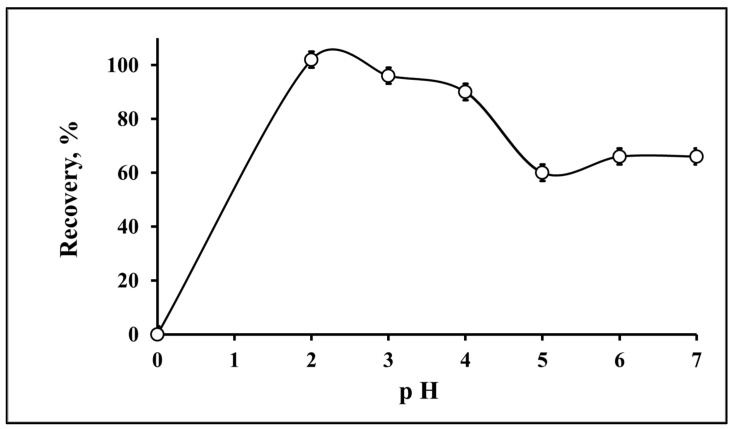
Evaluation of pH for the recovery of Pb(II) by DES-ME (*n* = 3).

**Figure 2 molecules-25-04794-f002:**
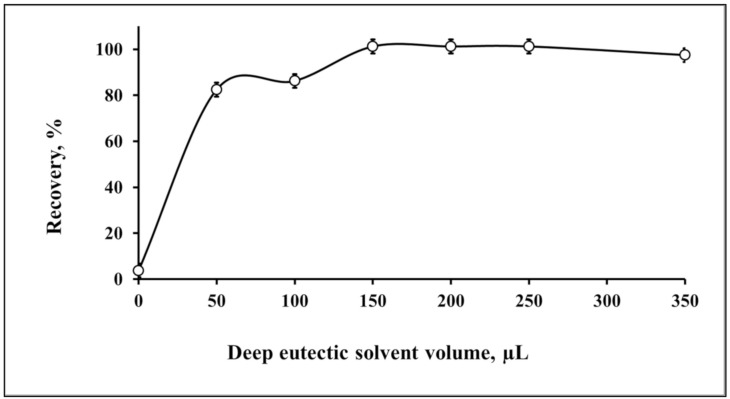
Optimization of the amount of the deep eutectic solvent for the recovery of Pb(II) by microextraction (*n* = 3).

**Figure 3 molecules-25-04794-f003:**
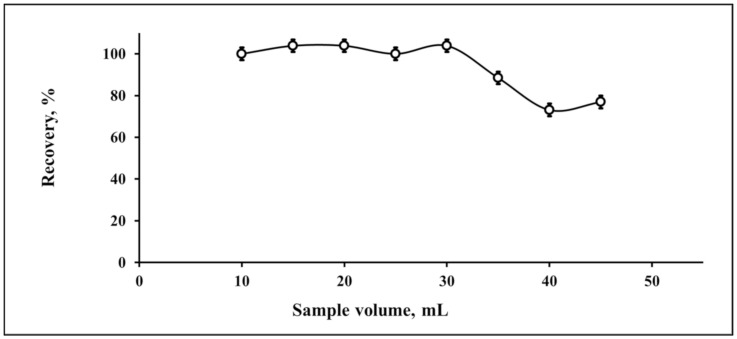
Evaluation of the sample volume for the deep eutectic-based microextraction of Pb(II) (*n* = 3).

**Table 1 molecules-25-04794-t001:** Recoveries of Pb(II) microextraction by deep eutectic solvent under various matrix applications (*n* = 3).

Ions	Concentration (mg L^−1^)	Salts	Recovery (%)
F^−^	800	NaF	100 ± 1
PO_4_^3−^	500	Na_3_PO_4_	95 ± 2
Ni^2+^	5	Ni (NO_3_)_2_·6H_2_O	100 ± 2
NO_3_^−^	500	NaNO_3_	103 ± 2
Co^2+^	10	Co(NO_3_)_2_	95 ± 2
Zn^2+^	10	Zn(NO_3_)_2_	102 ± 2
CO_3_^2−^	1200	Na_2_CO_3_	95 ± 3
K^+^/Cl^−^	5000	KCl	102 ± 2
Mg^2+^	1500	Mg(NO_3_)_2_·6H_2_O	98 ± 4
Ca^2+^	800	CaCl_2_	103 ± 2
SO_4_^2−^	400	Na_2_SO_4_	97 ± 1
Fe^2+^	10	Fe(NO_3_)_2_·9H_2_O	94 ± 3

**Table 2 molecules-25-04794-t002:** Validation of the DES-ME of Pb(II) from the certified reference material (*n* = 3).

Reference Material	Certified Value in Last Phase	Found Value, mg L^−1^	Recovery (%)
TMDA 64.2	1.4	1.43 ± 0.0	100 ± 0.0
TMDA 53.3	1.7	1.65 ± 0.02	97 ± 1.2
NCSDC-73349	1.4	1.43 ± 0.02	102 ± 1.4

**Table 3 molecules-25-04794-t003:** Determination of Pb(II) in samples from water and food extracts (*n* = 3).

Real Sample		Pb(II) Concentration Mean ± Standard Deviation
Water samples (mg L^−1^)	Tap water	BLD *
	Lake water	0.15 ± 0.02
	River water	0.13 ± 0.01
Food samples (mg kg^−1^)	Salted peanuts	0.1 ± 0.02
	Chickpeas	BLD
	Roasted yellow corn	0.08 ± 0.01
	Pistachios	BLD *
	Almonds	BLD *

* Below the limits of detection.

**Table 4 molecules-25-04794-t004:** Addition/recovery of Pb(II) from water and food samples using the developed deep eutectic microextraction.

Sample Matric	Spiking Pb(II) 0.5 mg L^−1^		Spiking Pb(II) 1.0 mg L^−1^		Spiking Pb(II) 2.0 mg L^−1^	
	Detected Pb(II) (mg/L)	Recovery	Detected Pb(II) (mg/L)	Recovery	Detected Pb(II) (mg/L)	Recovery
Tap water	0.48 ± 0.10	95	0.99 ± 0.00	99	1.95 ± 0.30	97
Chickpeas	0.50 ± 0.02	100	1.01 ± 0.10	101	2.02 ± 0.01	101
Pistachios	0.48 ± 0.03	95	0.92 ± 0.01	92	1.95 ± 0.01	97.4

**Table 5 molecules-25-04794-t005:** A comparison of the developed DES-ME of Pb(II) with microextraction methods from the literature.

Extraction Procedures	LODs (μg L^−1^)	Reference
Solid-phase extraction	0.1–2.8	[21]
Fe_3_O_4_ nanoparticles and ultrasound assisted dispersive liquid–liquid Microextraction (NPU-DLLME)	5.7	[23]
Dispersive liquid–liquid microextraction (DLLME)	0.00095	[24]
Dispersive liquid–liquid microextraction (DLLME)	0.000072	[30]
Displacement-dispersive liquid–liquid microextraction by solidification of floating organic	0.7	[32]
In-syringe system using ionic liquid for microextraction	0.281	[33]
Deep eutectic and air-assisted liquid microextraction	0.00060	[41]
DES-ME	0.008	This study

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
