# Peer review of "Deep Eutectic Solvent-Based Microextraction of Lead(II) Traces from Water and Aqueous Extracts before FAAS Measurements"

_molecules, 2020, doi:10.3390/molecules25204794_

Round 1
Reviewer 1 Report
The manuscript "Deep eutectic solvent based microextraction of lead(II) traces from water and aqueous extracts before FAAS measurements" present the results of study conducted in order to develop a microextraction process by applying deep eutectic solvents mixture composed from α-Benzoin oxime and iron(III) chloride dissolved in phenol for attraction and separation of Pb(II) from aqueous samples.
The manuscript can contribute to monitoring of toxic lead(II) presence in environment.
Therefore I suggest acceptance of this manuscript with minor revision since I found several parts of text which should be corrected (see attachment).

Author Response
Response:
Thanks for your comments and suggestions to improve our manuscript. All suggestions and correction have been done in the manuscript, and marked with red text and yellow background.
Reviewer 2 Report
The efficiency of the micro-extraction is impressive and this has significant impact on analysis of Pb in the environment. However the manuscript has many significant problems that require very major changes before it can be accepted. I will list them briefly in the following: 1) The paper is written in a very poor language. This has a negative effect on understanding the discussion and introduction. Therefore the paper should be rewritten with a very good attention to English writing, and fixing so many grammatical mistakes and typos. 2) The introduction is not clear. The authors need to describe why the micro-extraction is appropriate and later in the discussion they should do a better job comparing their results and the practicality of their method with the methods used currently for trace analysis of Pb. Also in the introduction the authors need to add citations for different statements that they have not justified. In addition they need to cite some reviews on DES. 3) In general in many places the authors used abbreviations without any definition. They should fix this. 4) Probably the most major problem is lack of molecular discussion of their results. This makes the paper more like a lab report with no insight! The authors should address this and this may require a significant change and restructuring of the paper. 5) I think the reported uncertainties in tables are not realistic considering the recoveries reported such as 106 percent! I strongly recommend that the authors would re-evaluate their uncertainties everywhere in the paper. 6) Add details about all material used. Just listing them is not enough! Some characterization of the DES is needed. How the pH did get adjusted? Add those details. Details like this are missing! For these reasons I suggest major revision although no control is missing.Author Response
Thanks for reviewing our manuscript. We have responded to all of them as follow:
All changes are marked with red color text:
1-The paper is written in a very poor language. This has a negative effect on understanding the discussion and introduction. Therefore the paper should be rewritten with a very good attention to English writing, and fixing so many grammatical mistakes and typos.
Response:
The English language is revised with an efficient English language professor.
2-The introduction is not clear. The authors need to describe why the micro-extraction is appropriate and later in the discussion they should do a better job comparing their results and the practicality of their method with the methods used currently for trace analysis of Pb. Also in the introduction the authors need to add citations for different statements that they have not justified. In addition they need to cite some reviews on DES.
Response:
The introduction part is revised. Additional description about the importance of the microextraction process for preconcentration and accurate analysis are added and some new references is discussed and cited.
- Jalili, V.; Barkhordari, A.; Norouzian Baghani, A. The role of microextraction techniques in occupational exposure assessment. A review. J. 2019, 150, 104086.
- Yamini, Y.; Rezazadeh, M.; Seidi, S. Liquid-phase microextraction – The different principles and configurations. TrAC - Trends Anal. Chem. 2019, 112, 264–272.
- Saqaf Jagirani, M.; Soylak, M. A Review: Recent Advances in Solid Phase Microextraction of Toxic Pollutants Using Nanotechnology Scenario. J. 2020, 105436.
- Rutkowska, M.; Płotka-Wasylka, J.; Sajid, M.; Andruch, V. Liquid–phase microextraction: A review of reviews. J. 2019, 149, 103989.
In addition, a table of comparison is added (Table 5) for comparison of the efficiency of the developed deep eutectic solvent base microextraction with other methods from the literature.
Furthermore, more discussion and reviews about the deep eutectic solvents are added to the manuscript.
- Smith, E.L.; Abbott, A.P.; Ryder, K.S. Deep Eutectic Solvents (DESs) and Their Applications. Rev. 2014, 114, 11060–11082.
- Kalhor, P.; Ghandi, K. Deep eutectic solvents for pretreatment, extraction, and catalysis of biomass and food waste. Molecules 2019, 24.
- Abolghasemi, M.M.; Piryaei, M.; Imani, R.M. Deep eutectic solvents as extraction phase in head-space single-drop microextraction for determination of pesticides in fruit juice and vegetable samples. J. 2020, 158, 105041.
- Sorouraddin, S.M.; Farajzadeh, M.A.; Okhravi, T. Application of deep eutectic solvent as a disperser in reversed-phase dispersive liquid-liquid microextraction for the extraction of Cd(II) and Zn(II) ions from oil samples. Food Compos. Anal. 2020, 93, 103590.
- Zounr, R.A.; Tuzen, M.; Khuhawar, M.Y. A simple and green deep eutectic solvent based air assisted liquid phase microextraction for separation, preconcentration and determination of lead in water and food samples by graphite furnace atomic absorption spectrometry. Mol. Liq. 2018, 259, 220–226.
3-In general in many places the authors used abbreviations without any definition. They should fix this.
Response:
The definitions are added to the abbreviations in the manuscript.
4-Probably the most major problem is lack of molecular discussion of their results. This makes the paper more like a lab report with no insight! The authors should address this and this may require a significant change and restructuring of the paper.
Response:
The discussion part is improved by adding more explanation about separation and microextraction of lead by the deep eutectic solvents.
5-I think the reported uncertainties in tables are not realistic considering the recoveries reported such as 106 percent! I strongly recommend that the authors would re-evaluate their uncertainties everywhere in the paper.
Response:
The recoveries in all tables are revised.
6-Add details about all material used. Just listing them is not enough! Some characterization of the DES is needed. How the pH did get adjusted? Add those details.
Response:
Details about the material sources and pH adjustment are added to the materials and methods part.